# Yield and Fruit Properties of Husk Tomato (*Physalis phyladelphica*) Cultivars Grown in the Open Field in the South of West Siberia

**Natalia Naumova** [1,*] **, Taisia Nechaeva** [1]**, Oleg Savenkov** [1] **and Yury Fotev** [2]

[1] Institute of Soil Science and Agrochemistry SB RAS, Novosibirsk 630090, Russia; nechaeva@issa-siberia.ru (T.N.); savenkov@issa-siberia.ru (O.S.)

[2] Central Siberian Botanical Garden SB RAS, Novosibirsk 630090, Russia; fotev_2009@ngs.ru

* Correspondence: naumova@issa-siberia.ru

**Abstract:** Husk tomato (*Physalis philadelphica* Lam.) a source of functional food and medicinal compounds, has attracted renewed interest for production in temperate zones. Field-grown husk tomato yield and fruit properties and their relationship with soil chemistry and temperature were studied in the south of West Siberia, Russia, at five experimental sites. At each site, a microplot experiment with two cultivars was conducted. Basic soil chemical properties and fruit pH and dry matter, total carbon, nitrogen, and ascorbic acid content were determined. Both cultivars grew and yielded very well, producing on average 70 fruits, or 1.46 kg, per plant, with 14 mg ascorbic acid per 100 g fresh weight, 9.0% dry matter, and juice pH of 4.1. Variation in environmental conditions among sites was the major factor determining production and fruit property variation, with cultivar biology accounting for 10%. The cultivars responded differently to some soil properties, but generally their yield and fruit quality depended on soil pH and labile phosphorous and potassium. Thus, husk tomato has remarkable capacity for vigorous yields in unprotected conditions in West Siberia, despite air and soil temperatures that are much lower than in its region of origin. Detailed studies are needed to elucidate its response to varying solar radiation and atmospheric precipitation.

**Keywords:** husk tomato; *Physalis philadelphica* Lam.; soil chemical properties; fruit quality; ascorbic acid; open field experiment; West Siberia; North Asia

## 1. Introduction

Husk tomato (*Physalis philadelphica* Lam. (synonymic *Physalis ixocarpa* Brot.)), also known as Mexican husk tomato, or tomatillo, is the most widely distributed species of *Physalis*. It is not only an interesting vegetable crop with a wide ecological niche for growing it [1], producing plenty of fruits, but it is also a medicinal plant with a long history of enthnomedicinal use [2,3]. Recently, husk tomato has been enjoying increasing attention in many countries, including Russia, due to new information on its chemical composition [4–6], health benefits, etc. [7–11]. Over the last few decades in Russia, the crop has been grown only by small farmers and individual gardeners; however, the last century saw quite a lot of attention paid to husk tomato. It was introduced into Russia in 1926, and was vigorously studied between 1920 and the 1950s [12], when field experiments were set up at experimental stations throughout the entire country [13]. Industrial production was started in the country in the Far East, where several thousands of hectares were occupied by husk tomato in the 1950s [14].

Currently such countries as the United States, Mexico, Canada, and Spain are the main players in the wholesale tomatillo market [15]. However, in Russia, there is no wide-scale production of husk tomato, and it only receives attention from small farmers and individual gardeners; so, the crop at present has little economic significance nationwide.

Husk tomato as a crop for functional nutrition and novel medicinal use together with global warming have resulted in increased interest in crop performance assessment in areas previously not involved in its cultivation. In the south of West Siberia (55–60° N, 59–84° E), global climate change has been shown to manifest itself in significant positive trends in the growing season length. The latter, together with increased sums of growing degree days and precipitation [16], have resulted in a tendency for increased production across the south of West Siberia in the past several decades. The further anticipated increase in regional plant productivity [16] actualizes studies on the growth, development, and yields of more warm-climate-loving vegetable crops, including husk tomato, in the open field in the region.

Cultivars may react differently to changed environmental conditions, and their yield potential can be modified by many factors, including edaphic ones. Currently, little is known about husk tomato production and possible cultivar differences in the open field in West Siberia. In addition, there are no data about biological production (i.e., biomass of major plant components, including those that are non-consumable by humans) of husk tomato. However, the recent intensification of studies on non-consumable aboveground and/or belowground components of various crops have increased the amount of attention paid to husk tomato in this respect as well.

The aim of this work was to study husk tomato yield and fruit properties and their relationship with soil chemistry and temperature in open field production in the south of West Siberia, Russia, during one growing season at five experimental sites with varying soil and environmental conditions.

## 2. Materials and Methods

### 2.1. Experimental Sites

To study the quantity and quality of husk tomato biological production, as well as fruit yield and properties, microplot field experiments were carried out at five experimental stations during the 2016 growing season in the forest-steppe zone in West Siberia (Figure 1).

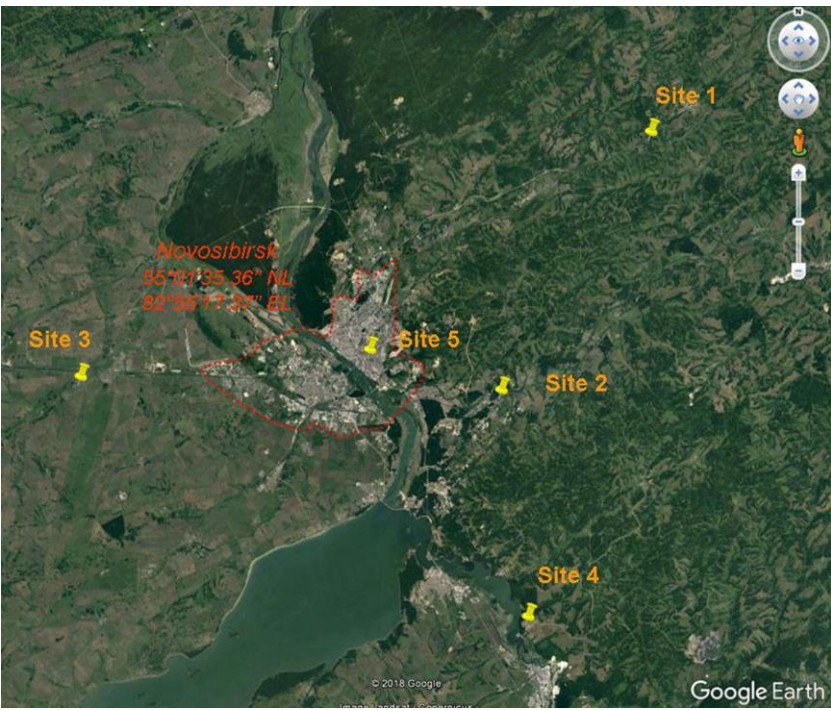

**Figure 1.** The location of the study sites.



The climate of the region is classified as sharply continental, with the average (June, July, August) maximal temperature in summer ranging from 22 to 26 °C and the average precipitation ranging from 40 to 65 mm per month, with a 119-day frost-free period. At each experimental station, air (2 m above the soil surface) and soil (at a depth of 1, 10, and 20 cm) temperatures were registered by Thermochron DS1921G data loggers (iButtonLink, USA), and the respective temperature sums were calculated for the duration of the experiment (103 days) (Table 1).

**Table 1.** The geographical location of experimental sites and their temperature sums (°C·day) over the growing period.

|  | Site 1 | Site 2 | Site 3 | Site 4 | Site 5 |
|---|---|---|---|---|---|
| Northern latitude | 55°15′39.71″ | 54°57′55.31″ | 54°58′57.90″ | 54°42′21.10″ | 55°00′46.04″ |
| Eastern longitude | 83°31′42.12″ | 83°13′09.92″ | 82°22′43.00″ | 83°16′02.60″ | 82°57′27.58″ |
| | | | Temperature sums | | |
| Air | 1403 | 1448 | 1485 | 1458 | 1570 |
| Soil, 1-cm depth | 1544 | 1357 | 1403 | 1517 | 1404 |
| Soil, 10-cm depth | 1387 | 1325 | 1338 | 1536 | 1381 |
| Soil, 20-cm depth | 1458 | 1256 | 1257 | 1587 | 1314 |

Experiments were conducted on loamy agricultural soils, common to the region, classified as Luvic Greyzemic Phaeozems (Siltic, Aric) according to the World Reference Base for Soil Resources [17]. Overall, the experimental plots had rather high soil organic carbon content and slightly acidic pH, which are favourable for plant growth and development (Table 2). Overall, the diversity of the soil properties at the experimental stations where the microplot field experiments were performed allows us to extend the conclusions over a wider gradient of soil and environmental conditions.

**Table 2.** The soil chemical properties at different sites before the start of the microplot field experiments.

|  | Site 1 | Site 2 | Site 3 | Site 4 | Site 5 |
|---|---|---|---|---|---|
| $pH_{H2O}$ | 5.77 | 5.91 | 5.96 | 6.01 | 5.65 |
| Soil organic carbon, % | 6.7 | 12.3 | 11.4 | 3.2 | 4.0 |
| $N-NH_4$, mg/kg | 1.5 | 3.0 | 2.2 | 2.8 | 2.3 |
| $N-NO_3$, mg/kg | 11 | 12 | 56 | 12 | 29 |
| P, mg/kg | 2.0 | 11.2 | 2.7 | 4.6 | 18.7 |
| Na, mg/kg | 150 | 117 | 105 | 16 | 14 |
| K, mg/kg | 905 | 342 | 315 | 342 | 362 |
| Mg, mg/kg | 519 | 776 | 652 | 101 | 200 |
| Ca, g/kg | 4.2 | 6.0 | 5.0 | 2.9 | 1.5 |

### 2.2. Experimental Setup

Two cultivars of *Physalis philadelphica* Lam., 'Konditer' (Confectioner) and 'Slivoviy Jam' (Plum Jam), were used. Confectioner has been listed in the Russian State Crop Register since 1990, being recommended for most of the regions of the country. The seeds for the study were provided by the seed bank of the Central Siberian Botanical Garden SB RAS (Novosibirsk, Russia). The plants have rounded/flat-rounded shaped fruits that are light green during the unripe stage and light yellow when ripe. Plum Jam, although not registered, has its seeds distributed by the Russky Ogorod seed-producing and trading company. The cultivar produces rounded/flat-rounded fruits, the colour of which becomes purple-violet when they reach biological ripeness.

Seedlings were grown in cassettes (7 × 7 cm) in a peat substrate in the Central Siberian Botanical Garden SB RAS (Novosibirsk, Russia). The young plants were planted out on June 10, 2016, at the age of 50 days into open-field microplots at a density of one plant per 0.25 m². No fertilizers were applied. At each experimental station, the experimental setup was similar, with two cultivars and four replicates, randomized, so that, overall, there were eight plants/microplot at each of the five experimental stations.

### 2.3. Biomass Collection and Analyses

The growing season in the open field in West Siberia is short, with cool nights already occurring in August, which can prevent the majority of fruits from ripening *in situ*. So, husk tomato fruits were collected repeatedly during the growing season, starting at the end of July, as soon as they stopped increasing in size and reached color maturity. At the end of the experiment, all consumable fruits were collected. The ripe husk tomato fruits are shown in Figure 2. Above- and belowground biomass were also determined at the end of the experiment, just prior to the first nighttime frosts in the middle of September. Physical and chemical properties of the fruit including dry matter content, organic carbon, total nitrogen, pH, and ascorbic acid content were quantified by standard techniques. After harvest, dry matter and ascorbic acid content and juice pH were assessed. Dry matter content was determined in chopped samples from each plant by drying in an oven at 80 °C until a constant weight was achieved. Total carbon and nitrogen content were determined by a CHN 2400 Series II elemental analyzer (Perkin Elmer, USA). Ascorbic acid was measured by the standard iodine titration technique as described in [18] and was expressed on a fresh mass basis.

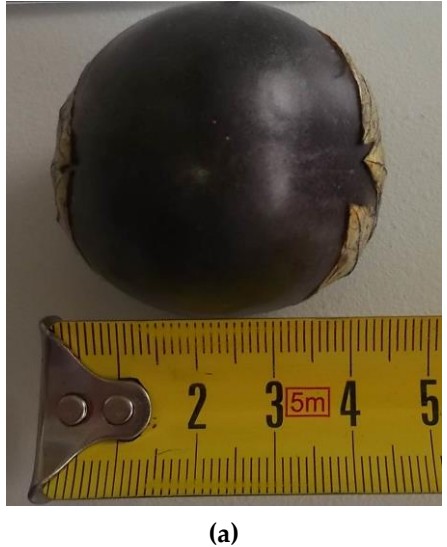

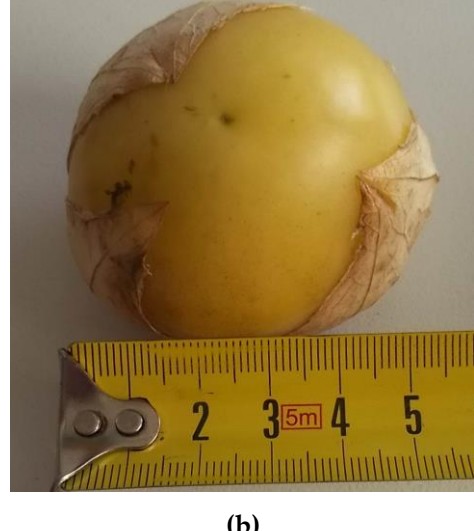

(a) (b)

**Figure 2.** The ripe fruits of the studied husk tomato cultivars: Plum Jam (**a**) and Confectioner (**b**).

### 2.4. Soil Sampling and Analyses

Soil was sampled before the start of the experiment (June 2016). At each experimental microplot, i.e., from under each plant, three subcores were taken from a 0–20-cm soil layer and bulked together to comprise one composite sample. Field-moist soil samples were sieved through 2-mm mesh and stored in a refrigerator (+4 °C) before analysis. The content of soil organic carbon (SOC) was estimated by stepwise loss on the ignition method [19] using 2–4 g soil aliquots, and multiplying the loss on ignition by 0.58. For these analyses, soil was air-dried. Available forms of macronutrients ($NO_3^-$, $NH_4^+$, $P_2O_5$) were determined in field-moist samples by standard techniques. Briefly, nitrate and ammonium were measured colourimetrically in 2N KCl extracts [20], and available P was extracted with 0.5 M $NaHCO_3$ solution and determined colourimetrically [21]. Soil pH was measured in a supernatant of

soil–water solution (1:5 v/v) [22] by a PB-11 pH meter (Sartorius, Germany). Exchangeable $K^+$, $Na^+$, $Ca^{2+}$, and $Mg^{2+}$ were determined in ammonium acetate extracts as described in [23].

*2.5. Statistical Analysis*

The data were analyzed by descriptive statistics, ANOVA, and a correlation analysis using the *Statistica 13.3* software package, and principal component analysis (PCA) using the PAST software package [24]. Correlation coefficients and factor effects were considered statistically significant at the $P \leq 0.05$ level.

## 3. Results

*Husk Tomato Yield*

At each experimental site, husk tomato plants grew vigorously and produced fruit with no apparent limitations or diseases (Table 3). Averaged over experimental sites, the husk tomato fruit yields were practically the same for both cultivars, averaging 1.46 kg (fresh mass) per plant. As one plant grew on 0.25 $m^2$, this yield was equivalent to 5.84 kg/$m^2$.

**Table 3.** Some characteristics of husk tomato production averaged over five experimental sites (mean ± standard deviation).

| Parameter | Plum Jam | Confectioner | *P*-Value* |
|---|---|---|---|
| Number of fruit per plant | 79.6 ± 35.0 | 60.5 ± 32.2 | 0.12 |
| Fruit yield per plant, g * (F) | 1426 ± 669 | 1502 ± 1190 | 0.83 |
| Maximal fruit mass, g ** | 33 ± 10 | 38 ± 19 | 0.36 |
| Mean fruit mass, g * | 17 ± 4 | 23 ± 8 | 0.03 |
| Aboveground biomass, g * (AG) | 1001 ± 560 | 1139 ± 734 | 0.56 |
| Belowground biomass, g * (R) | 86 ± 48 | 76 ± 46 | 0.57 |
| Ratio AG/R | 12.0 ± 4.6 | 14.9 ± 3.2 | 0.05 |
| Ratio AG/F | 0.8 ± 0.4 | 1.2 ± 0.9 | 0.11 |

* as estimated by ANOVA. ** fresh mass. PJ, Plum Jam; C, Confectioner; F, fruit yield per plant; AG, aboveground biomass; R, belowground biomass.

However, the husk tomato cultivars differed significantly (by almost 2-fold) in the ratio of the aboveground biomass to fruit mass, thus evidencing the higher plant energy expenditures by Confectioner for fruit production as compared to that of PJ.

There was no difference in fruit quality between the cultivars (Table 4).

**Table 4.** Some chemical properties of husk tomato fruits as averaged over five experimental sites (mean ± standard deviation).

| Parameters | Plum Jam | Confectioner | *P*-Value * |
|---|---|---|---|
| pH | 4.0 ± 0.3 | 4.2 ± 0.4 | 0.10 |
| Dry matter, % | 8.7 ± 1.2 | 9.3 ± 1.2 | 0.23 |
| $C_{tot}$, % ** | 44.9 ± 3.3 | 44.1 ± 2.7 | 0.51 |
| $N_{tot}$, % ** | 2.2 ± 0.7 | 2.1 ± 0.7 | 0.63 |
| C/N (atomic) | 27 ± 8 | 28 ± 9 | 0.79 |
| Ascorbic acid, mg/100 g *** | 14.1 ± 6.4 | 13.4 ± 4.5 | 0.76 |

* as estimated by ANOVA, **dry mass basis, *** fresh mass basis.

The ANOVA results for aboveground husk tomato characteristics revealed that the major part of the data variance was due to the experimental site effect (Table 5). The experimental site effect embraced soil and weather (solar radiation, precipitation, etc.) conditions for plant growth and development. The correlation analysis showed some statistically significant correlation coefficients between husk tomato production characteristics and soil chemical and temperature properties

(Figure 4). Confectioner production characteristics, such as the number and mass of fruits per plant, aboveground biomass, and fruit mean mass, were negatively correlated with soil pH. The Confectioner fruit yield per plant and fruit mean mass were positively correlated with exchangeable K in soil. The same was true for Plum Jam as well. The aboveground biomass of both cultivars was positively correlated with soil labile P. The ratio of above- to belowground biomass for both cultivars was negatively correlated with basic anions, such as $Na^+$, $Ca^{++}$, and $Mg^{++}$; the correlation was statistically significant for Plum Jam.

**Table 5.** The ANOVA results for husk tomato production and fruit properties: the contribution of factors (in %) to the total variance and the probability of a null hypothesis (in brackets).

| Particulars | Factor | | |
|---|---|---|---|
| | Site (A) | Cultivar (B) | A × B |
| Production properties | | | |
| Number of fruits | 32 (0.03) | 10 (0.05) | 7 (0.58) |
| Fruit yield (F) | 44 (0.00) | 1 (0.87) | 13 (0.19) |
| Mean fruit mass | 54 (0.03) | 7 (0.01) | 15 (0.01) |
| Aboveground biomass (AG) | 43 (0.00) | 1 (0.63) | 13 (0.19) |
| Belowground biomass (R) | 14 (0.32) | 2 (0.42) | 24 (0.10) |
| Ratio AG/R | 38 (0.00) | 13 (0.01) | 12 (0.18) |
| Ratio AG/F | 44 (0.00) | 17 (0.00) | 28 (0.00) |
| Fruit properties | | | |
| pH | 34 (0.02) | 1 (0.90) | 29 (0.03) |
| DM* | 26 (0.08) | 1 (0.85) | 27 (0.08) |
| $C_{tot}$ | 63 (0.00) | 2 (0.24) | 6 (0.49) |
| $N_{tot}$ | 76 (0.00) | 1 (0.99) | 5 (0.44) |
| C/N | 73 (0.00) | 1 (0.74) | 4 (0.62) |
| AA** | 3 (0.96) | 7 (0.26) | 22 (0.40) |

\* DM, dry matter content; \*\* AA, ascorbic acid content.

The total N content in husk tomato fruits of both cultivars, as well as the C/N stoichiometry, had similar correlations with the soil chemical and temperature properties, which is in contrast to those displayed by other fruit properties (Figure 5).

A principal component analysis of the data separated the cultivars by their production only in the plane of principal components 3 and 4, which together accounted for 23% of the data variance (Figure 3).

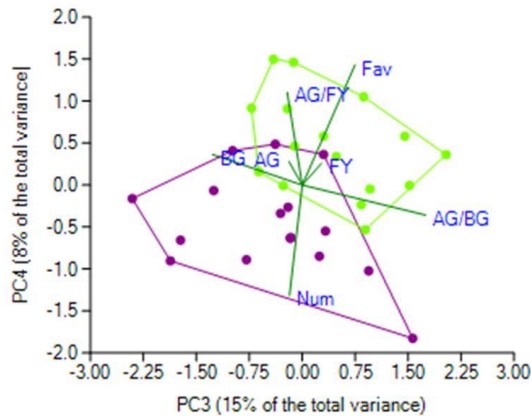

**Figure 3.** The location of husk tomato production characteristics (variables for analysis) and husk tomato cultivars in the plane of principle components 3 and 4. Abbreviations used for plant variables: AG, aboveground biomass; BG, belowground biomass; Num, the number of fruits; FY, fruit yield (mass); Fav, average mass per fruit; AG/FY, the ratio of the aboveground biomass to fruit yield and belowground biomass; AG/BG, the ratio of the above- to the belowground biomass. Violet circles represent Plum Jam, and green circles represent Confectioner.

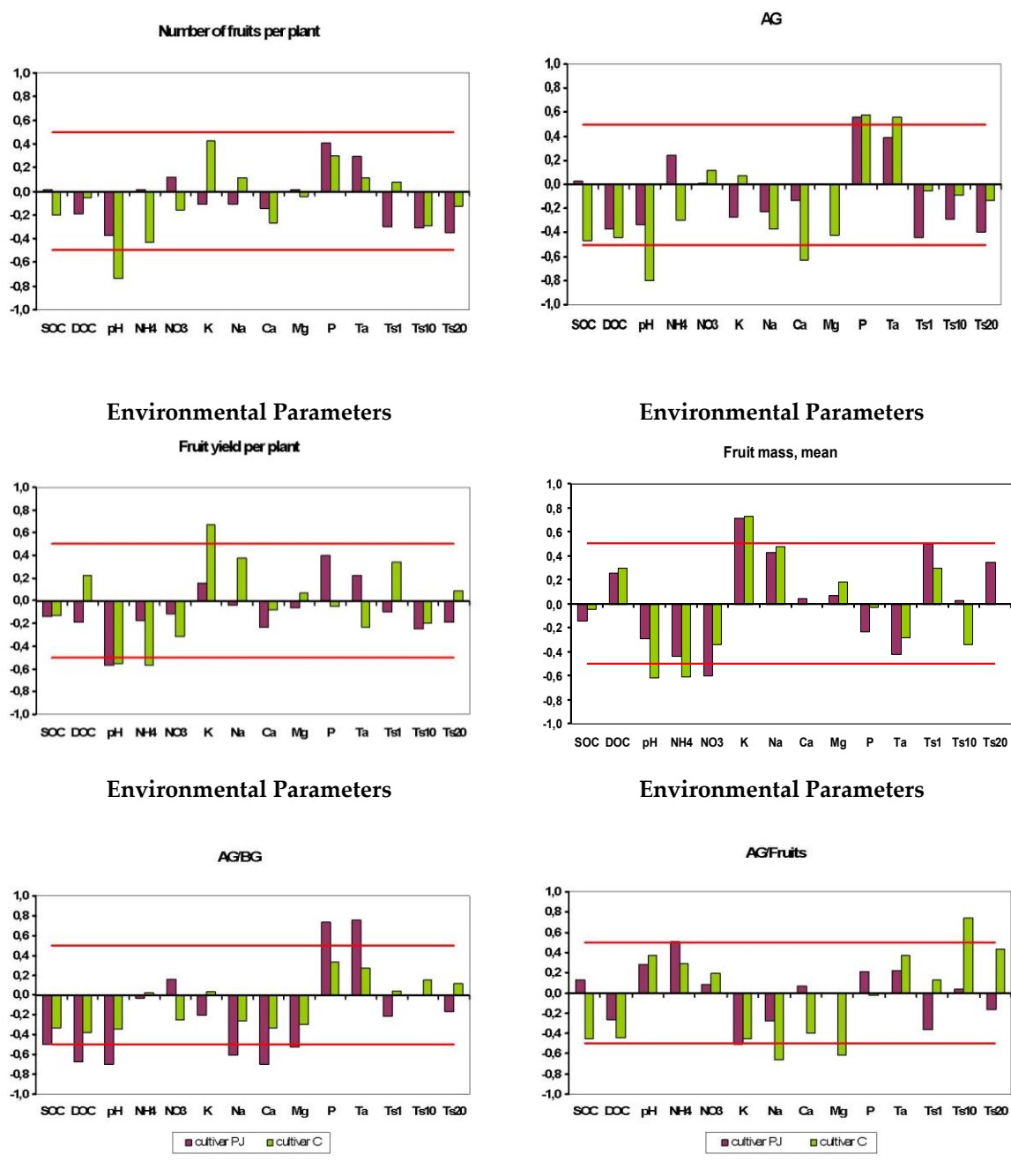

**Figure 4.** The correlation coefficients between Plum Jam (PJ) and Confectioner (C) husk tomato production characteristics, soil chemical properties, and soil and air temperature. Red lines indicate the levels of correlation coefficients, above or below which the correlation coefficients are statistically significant ($P \leq 0.05$). Environmental parameter abbreviations: SOC, soil organic carbon; DOC, dissolved organic carbon; $NH_4$ and $NO_3$, exchangeable soil ammonium and nitrate, respectively; Ta, the sum of air temperature over the growing season; Ts1, Ts10, and Ts20, the sum of soil temperatures at a depth of 1, 10, and 20 cm over the growing season, respectively.

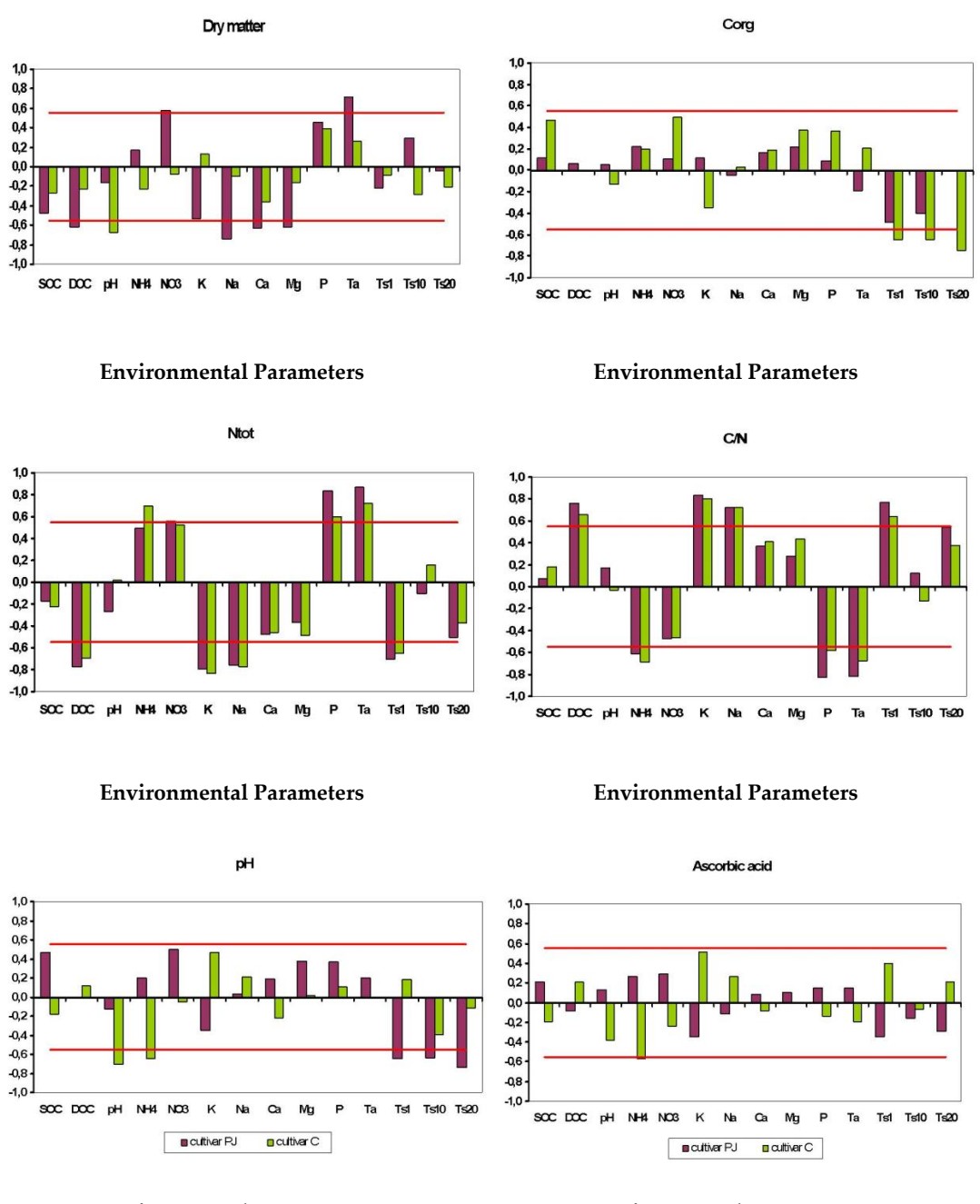

**Figure 5.** The correlation coefficients between Plum Jam (PJ) and Confectioner (C) husk tomato fruit characteristics, soil chemical properties, and soil and air temperature. Red lines indicate the values of correlation coefficients, above (or below) which correlation coefficients are statistically significant ($P \leq 0.05$). Environmental Parameter abbreviations: SOC, soil organic carbon; DOC, dissolved organic carbon; $NH_4$ and $NO_3$, exchangeable soil ammonium and nitrate, respectively; Ta, the sum of air temperature over the growing season; Ts1, Ts10, and Ts20, the sum of soil temperatures at a depth of 1, 10, and 20 cm over the growing season, respectively.

## 4. Discussion

The husk tomato fruit yield averaged 5.8 kg/m$^2$. This value is higher than the reported earlier for Russia; for example, some authors reported 3.0–4.5 kg/m$^2$ [14]. Commercial husk tomato varieties grown in an open field experiment yielded 5.6–6.4 kg/m$^2$ in New Hampshire, U.S.A. [25]; 1.1–1.9 kg/per plant (similar to our values) in Georgia, U.S.A. [26]; 1.1–2.6 kg/per plant in Mexico [27];

and 1.8 kg/per plant in the European part of Russia [28]. At the same time, husk tomato production in California was reported as ranging from 1.5 to 3.5 t/acre; i.e., 0.4–0.9 kg/m$^2$, which is extremely low and even seems to be erroneous [29]. Somewhat higher, but still low, yields of 2.0–2.4 kg/m$^2$ were reported for Florida [30]. Thus, we can conclude that the yield potential of both cultivars in the open field in the south of West Siberia even without fertilization is quite high and comparable with the region of its origin.

Data are scarce on whole plant development of the husk tomato. The values of the aboveground biomass reported here are twice as low as the values of the vegetative fresh mass of tomato husk plants (2.1–2.5 kg/per plant) grown in Georgia, U.S.A. [26]. However, there the ratio of the vegetative fresh mass to fruit yield was more than 2-fold higher. Thus, our cultivars seemed to be more yield-efficient; i.e., required less photosynthetic biomass for fruit production. As for the ratio of the above- and belowground biomass, it was similar to the one reported for Georgia, U.S.A. [26]. There, the production of non-consumable husk tomato biomass averaged 4.3 kg/m$^2$ for the aboveground biomass and 0.3 kg/m$^2$ for the belowground biomass, providing an important source for (vermi)composting and other uses.

The temperature of the soil layers, where husk tomato develops its roots, is very important for plant growth and development. However, in our study we did not find a statistically significant correlation between soil layer temperatures and husk tomato growth and development, which may be due to the wide range of root zone temperatures that are not stressful for the plant [26], as well as other factors masking a possible root zone temperature effect. However, the cultivar Plum Jam had a negative correlation between its root zone temperature and the fruit organic carbon content.

The fruit properties of Confectioner were practically the same as reported recently for the husk tomatofruits grown in the European part of Russia, albeit in protected conditions [31]; however, the vitamin C content was higher in our study. The latter was also higher as compared, for example, with the vitamin C content of raw tomatillo fruits in the USDA National Nutrient Database for Standard Reference [32], and much higher than the values of 5.5–7.0 mg/100 g fresh fruit mass, reported earlier [27], for fruits of the di- and tetraploid husk tomatoes grown in Mexico. However, the same *Physalis* species, but a native one, grown in Brazil, was recently reported to have 26 mg of ascorbic acid per 100 g fresh mass [33], which is twice as high as the values in our study. Compared to husk tomato fruit grown in India, our values are close to their lower values [5]. It should be noted that, in our study, husk tomato fruits were collected at the stage of technical maturity and then ripened during storage at 22 °C. The results show that at least some characteristics of fruit nutritional quality were not compromised by such harvesting; however, there is no doubt that nutritional quality is at its highest in *in situ* ripened mature fruits. So, the ascorbic acid synthesis/accumulation performance of husk tomato in our experiment can be considered rather good, as four to five mean-sized husk tomato fruits, obtained in our study, provide 60 mg, or 73%, of the recommended daily dose of vitamin C [34].

It should be noted that the field experiment setup of two cultivars and four replicated microplots was replicated at five study sites, or localities, to provide a broader gradient of environmental conditions that cannot be controlled in the open field. Among such conditions, photosynthetically active solar radiation and natural precipitation are the most important for crop growth and development. The study sites were located within a quadrate of *ca.* 65 km × 80 km, and the soils at the study sites, despite being different in some properties, all fell within the range that is generally favourable for crop production. The combination of all these factors resulted in the predominance of the site effect on the variance of tomato husk plant production and fruit properties. The correlation analyses revealed few statistically significant relationships between plant production and fruit characteristics with soil properties. In our view, this may due to variation in photosynthetically active radiation among the sites, and could have been the major force driving between-site variation in plant production and fruit properties. Such field experiment designs, where yearly (temporal) replication is substituted by a spatial one, are rather rare. Our data agree with the findings of Mexican researchers who found that the effect of localities, i.e., study sites, accounted for most of the variance in husk tomato fruit yield,

average fruit mass and size, as well as fruit pH and total soluble solids [27]. Interestingly, in their study, the effect of location accounted for just 2% of ascorbic acid variance in fruits. Here, we report 3%, which implies that the synthesis and accumulation of this vitamin in husk tomato fruits is governed by a complex interplay of environmental factors. The same complex interplay of environmental factors, inherent in our experimental scheme, might account for the fact that a statistically significant correlation could not be found with certain variables presumably meaningful for plant production and fruit chemistry properties, such as organic matter. However, the cases where such a correlation was found stress the importance of soil properties. The positive correlation between aboveground biomass of both cultivars and soil labile P indicated that phosphorus may be a limiting factor for husk tomato plant growth on these soils without mineral fertilization. In our study, we also found a negative correlation between soil pH and some production characteristics of husk tomato, especially for the cultivar Confectioner, even within the narrow between-site pH range of 0.36. Thus, we may conclude that husk tomato is a crop that is sensitive to soil pH. The positive correlation between soil exchangeable potassium, fruit mean mass, and fruit yield per plant reiterates the importance of the nutrient, known for its role in plant polymer synthesis, for tomato husk growth and production.

In our study, the effect of cultivar on above- and belowground biomass, as well as fruit yield, turned out to be negligible, both statistically and agriculturally. However, the cultivar effect was quite pronounced in the number of fruits per plant, the mean fruit mass, and the ratios of aboveground to belowground biomass and of the former to the fruit mass. Such a between-cultivar difference over the range of environmental factors implied tight genetic control of these plant characteristics in each cultivar. A similar pattern of correlation between total and C/N in fruits with soil chemical and temperature properties might indicate the fact that such a basic chemical property of the species is very tightly controlled genetically.

## 5. Conclusions

This one-year field trial showed that both cultivars of husk tomato have more than sufficient potential for vigorous development and yielding in the open field in West Siberia under its current climate. Husk tomato can successfully grow in the south of West Siberia under a range of environmental conditions, not being susceptible to air and soil temperature sums much lower than in its region of origin, i.e., Mexico. The tested cultivars responded differently to some soil properties. Their yield and fruit quality depended on pH, potassium, and contents of other alkaline elements. Detailed studies are needed to elucidate husk tomato response under unprotected conditions in the open field to varying solar radiation, as well as to other environmental conditions, such as atmospheric precipitation.

**Author Contributions:** Author N.N. designed the study, performed the statistical analyses, and wrote the first draft of the manuscript. All authors carried out field experiments. Author Y.F. supplied plant material. All authors read and approved the final manuscript.

**Funding:** This research was funded by the Ministry of Science and Higher Education of the Russian Federation as the State Research Project VI.54.1.3.

**Acknowledgments:** The authors are very thankful to S.B. Drozdova, G.A. Bugrovskaya, and N.T. Vladimirova for their laboratory assistance.

**Conflicts of Interest:** The authors declare no conflict of interest.

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
