# Peer review of "Yield and Fruit Properties of Husk Tomato (Physalis phyladelphica) Cultivars Grown in the Open Field in the South of West Siberia"

_horticulturae, doi:10.3390/horticulturae5010019_

Round 1
Reviewer 1 Report
Reviewer comment manuscript Horticulturae-423324
Short summary
The study titled “Yield and fruit properties of husk tomato (Physalis phyladelphica) grown in the open field in the south of West Siberia” is focused on a one-year field study in micro plots to compare the adaptation to field conditions in a temperate climate atmospheric conditions of two cultivars of husk tomatoes. The main observation and conclusion is that husk tomatoes can be efficiently grown in these conditions. The study is within the scope of “Horticulturae” journal and can serve its readership well. However, it still needs some work to convey the information in a clearer manner.
General impression/comments
The manuscript and the study are worth publishing, yet it feels that the author should give more emphasis to the novelty of the study (e.g. write more within the global context and regarding climate change) on the one hand and on the other hand emphasize that it was a one-year study on a small scale.
G-1: The data can be presented in a clearer fashion to make it easier for the readers to follow. Some flow chart, general schema or a picture of the general design of the field study can be of help.
G-2: Be more aware that decisive conclusions from a one year study, such as: “that photosynthetically active radiation, varying among the sites, was the major force driving between-site variation in plant production and fruit” should be either avoided or tuned down.
Specific comments
Specific comments are according to the article structure
(i) Title
The title should mention that two cultivars were tested.
(ii) Abstract
Try to emphasize in the abstract what is new in this study and why it is important.
(iii) Introduction
General: the introduction should focus more on the present rather than on the past. It should also give more general details about husk tomato growing practice as this journal's readership will be more into understanding the agricultural part than into either health or history.
L32-L36: Too many words are devoted to things that were not in the core of this study, make it shorter.
L36-L51: As mentioned above, only a brief intro to the history should be enough, most of it can be omitted.
L51-L53: Expand to what is going on globally (as was done in the discussion, just in a more general way).
L58 – I would change “biological production” to “yield” or something similar.
(iv) Materials and methods
L69-71: How was the temperature taken/measured?
L74-79: The font or size changed.
L82-89: This description mostly belongs to the introduction and not to the materials and methods section. Just specify what are the cultivars and where did you purchase them.
L100-L101: Do not confuse between material and methods and results.
L104: How the pH was measured (instrument, technique)?
L109: Add a reference about the method.
(v) Results
L130: Change instead of “very well” to “no penalties or diseases observed” or something less arbitrary (not sure what “very well” means).
L137: Why this is significant? Is it statistically significant? If it is than give a p-value.
L149: Figure 1 c?
L167-L168: The results of the figure can be elaborated.
(vi) Discussion
L178:184: Please use unified MKS units for all the data.
L195: Add references.
L210 – L213: The font was changed.
L216: Add reference
L221: Add reference
L225-L229: Beware from strong claims, the soil is probably one of the most complex systems on earth (physically, chemically and biologically) and many other things can contribute to the differences such as (microbial populations, wind regiments, physical structure (not to be confused with soil characteristics and composition) the water flax, penetration and so on…).
L241-L244 What kind of sensitivity for lower pH, is there information about the optimal pH?
L247 – Add a reference
L252- L254: You didn’t test any genetic/molecular biology data, so I would omit this sentence.
(vii) Conclusions
Just make sure to emphasize that this is a one-year field trial.
(viii) References
Is ref 28 is empty?
(ix) Tables and Figures
Table 1: The temperature data can come as a bar graph and the geographic location can be added to the text.
Table 2: Geographic location is redundant. Are these results presenting multipole measures? If so can you provide a SD?
Table 3: What is R? You can call AG/F “Harvest index”.
Table 4: It seems that there is a significant difference of the pH. This is a very important characteristic for a food product.
Table 5: Can be omitted or transferred to a supplementary. If it stays, you should change the design to make it clearer.
Figure 4: If possible, change the color of the lines in the background so it would be easier to look at.

Author Response
Dear Reviewer,
Thank you very much for your thorough review and many useful comments, which we are addressing below.
General comments
Point 1 (G-1) The data can be presented in a clearer fashion to make it easier for the readers to follow. Some flow chart, general schema or a picture of the general design of the field study can be of help.
Response 1. Google map with study sites added. The scheme of microplots (8 quadrate microplots per site, with 4 micropolot per cultivar and 1 plant per microplot, randomized) seems to be sufficiently described in Section 2.2
Point 2 (G-2) Be more aware that decisive conclusions from a one year study, such as: “that photosynthetically active radiation, varying among the sites, was the major force driving between-site variation in plant production and fruit” should be either avoided or tuned down.
Response 2. Tuned down
Specific comments
Point 3 (i) Title
The title should mention that two cultivars were tested.
Response 3. Done
Point 4. (ii) Abstract
Try to emphasize in the abstract what is new in this study and why it is important
Response 4. We tried, but so far failed: t is difficult to change the abstract as it contains exactly 200 words
Point 5. (iii) Introduction
General: the introduction should focus more on the present rather than on the past. It should also give more general details about husk tomato growing practice as this journal's readership will be more into understanding the agricultural part than into either health or history.
Response 5. We found only some general info about husk tomato growing practice which can be summed as similar to tomato; research article usually describe rather specific aspects
Point 6. L32-L36: Too many words are devoted to things that were not in the core of this study, make it shorter.
Response 6. Shortened.
Point 7. L36-L51:As mentioned above, only a brief intro to the history should be enough, most of it can be omitted
Response 7. Mostly omitted
Point 8. L51-L53: Expand to what is going on globally (as was done in the discussion, just in a more general way)
Response 8. Done, but very briefly.
Point 9. L58: I would change “biological production” to “yield” or something similar
Response 9. We specified the meaning of the term “biological production” in the text, since it is not equivalent to “yield”.
Point 10. L69-71:How was the temperature taken/measured?
Response 10. The required specifics added.
Point 11. L74-79
The font or size changed
Response 11. Corrected.
Point 12. L82-89
Just specify what are the cultivars and where did you purchase them.
Response 12. Done
Point 13. L100-L101: Do not confuse between material and methods and results.
Response 13. Corrected
Point 14. L104: How the pH was measured (instrument, technique)?
Response 14. The required specifics added.
Point 15. L109: Add a reference about the method
Response 15. Added
Point 16. L130: Change instead of “very well” to “no penalties or diseases observed” or something less arbitrary (not sure what “very well” means)
Response 16. Changed
Point 17. L137: Why this is significant? Is it statistically significant? If it is than give a p-value.
Response 17. It is important as it pertains to the structure of the species/cultivar energy expenditures for fruit production, but it is not statistically significant (the p-values are supplied in Table 4).
Point 18. L149: Figure 1 c?
Response 18. Corrected
Point 19. L167-L168: The results of the figure can be elaborated.
Response 19. Yes, but then it will be mere speculation.
Point 20. L178:184: Please use unified MKS units for all the data.
Response 20. So changed
Point 21. L195, L216, L221, L247: Add reference
Response 21. Added or changed; however, as for line 221, we are afraid we do not understand the point that requires reference support, so could not amend anything.
Point 22. L225-L229: Beware from strong claims, the soil is…
Response 22. Tuned down
Point 23. L241-L244: What kind of sensitivity for lower pH, is there information about the optimal pH?
Response 23. We could not find information about the lowest possible pH for the crop. As for the optimal pH, we have not found any; besides, it would be very context-specific, so of what use?
Point 24. L252- L254: You didn’t test any genetic/molecular biology data, so I would omit this sentence
Response 24. Absolutely, but we wrote it on assumption that in the discussion section the author can put forward some explanatory ideas, or hypothesis, can not they?
Point 25. Conclusions: Just make sure to emphasize that this is a one-year field trial
Response 25. Done
Point 26. Is ref 28 is empty?
Response 26. Yes, it was a mistake; corrected.
Point 27. Table 1: The temperature data can come as a bar graph and the geographic location can be added to the text.
Response 27. It seems to us that a) transforming the geographical coordinates from the table into the text will come as cumbersome for the 5 sites; b) there are not much info about temperature sums, so we exact values might be of use for some researchers
Point 28. Table 2: Geographic location is redundant. Are these results presenting multipole measures? If so can you provide a SD?
Response 28. Coordinates removed from the table. We chose not to add sd as it will make the table cumbersome, serving no aim of the study.
Point 29. Table 3: What is R? You can call AG/F “Harvest index”.
Response 29. “R” stands for the belowground phytomass, i.e. roots; the table is adequately corrected.
Point 30. Table 4: It seems that there is a significant difference of the pH. This is a very important characteristic for a food product.
Response 30. It is, if we expand the level of statistical significance up to P≤0.10, as it is sometimes done in the ecological/environmental studies; p-values added to the Tables 3 and 4.
Point 31. Table 5
Can be omitted or transferred to a supplementary. If it stays, you should change the design to make it clearer.
Response 31. We would prefer for the table to stay in the main body of the manuscript; so changed the caption and the design to add clarity
Point 32. Figure 4: If possible, change the color of the lines in the background so it would be easier to look at.
Response 32. We do not mind any changes here, but we are afraid we do not understand why only Figure 4, and not Figure 3? Personally, we like the colours.

Reviewer 2 Report
The importance and economic significance of these cultivars can be better described.
The tables that are provided in the methods do not have full forms for the acronyms used (IL?).
The statistical analysis requires better description. Reps, plots, what was the level of significance
The figures have adequate descriptions in their headings but not the tables.These need to be improved.
Climatic differences between the differences of other growing regions to that of Siberia would add to the description and explain differences in yield.
A map of the field plot on google maps and plot design would be useful.
References for methods used need to be provided and are missing.
Author Response
Dear Reviewer,
Thank you very much for your thorough review and useful comments, which we are addressing below.
Point 1. The importance and economic significance of these cultivars can be better described.
Response 1. Some details added.
Point 2. The tables that are provided in the methods do not have full forms for the acronyms used (IL?).
Response 2. Corrected.
Point 3. The statistical analysis requires better description. Reps, plots, what was the level of significance.
Response 3. Some details added, among them p-values in Tables 3 and 4..
Point 4. The figures have adequate descriptions in their headings but not the tables.These need to be improved.
Response 4. Corrected according to our understanding of the problem.
Point 5. Climatic differences between the differences of other growing regions to that of Siberia would add to the description and explain differences in yield.
Response 5. We can only provide the weather data from the Novosibirsk meteostation, and it is located far from our experimental plots, we registered neither precipitation nor active solar radiation variation. The latter would have been useful, but we could not technically register it that year. Information about climatic differences, i.e. weather characteristics averaged over long periods (inRussiait is usually 30 years, i.e. currently for the period 1991-2010) could not explain any differences. Besides, Siberia is a huge region, as you well know, and climate comparison between regions will go far beyond the scope of the article, besides increasing the amount of information presented.
Point 6. A map of the field plot on google maps and plot design would be useful.
Response 6. The map added. But we do not see it as worthwhile to add a scheme of microplot experiment at each study site, as it is described in the “Experiment setup” section 2.2 and is rather simple, i.e. 8 microplots of 0.25 sq m, with one plant per microplot, 4 microplots per cultivar, randomized.
Point 7. References for methods used need to be provided and are missing.
Response 7. References added.

Round 2
Reviewer 1 Report
No more comments from my side